# Investigation on the Rheological Properties and Microscopic Characteristics of Graphene and SBR Composite Modified Asphalt

**Lijun Wang [1,\*], Fengxiang Liang [1], Zixia Li [2] and Qiang Zhao [1]**

[1] School of Civil Engineering and Transportation, Northeast Forestry University, Harbin 150040, China; lfx@nefu.edu.cn (F.L.); zqnefu@outlook.com (Q.Z.)
[2] State Grid Harbin Power Supply Company, Harbin 150000, China; zxialii@outlook.com
[\*] Correspondence: wlj@nefu.edu.cn; Tel.: +86-180-1950-6382

**Abstract:** Styrene-butadiene rubber (SBR) is commonly used as a modifier to enhance the low-temperature performance of asphalt. However, it is worth noting that while SBR modified asphalt exhibits good low-temperature performance, its high-temperature performance is comparatively inferior. This limitation significantly restricts the widespread use of SBR modified asphalt. As a new type of nanomaterial, graphene (GR) can change the microstructure of asphalt binder and provide asphalt with better mechanical, thermal, and adhesion properties. The main purpose of this study is to explore the influence of GR and SBR composite incorporation on the performance indexes of modified asphalt, and to study its compatibility and modification mechanism from the microscopic point of view of asphalt. The weight factor optimization system of modified asphalt was established by an analytic hierarchy process, and the optimum content of GR was determined to be 0.1% in a quantifiable way. The test results demonstrate that the inclusion of graphene substantially enhances the high-temperature rutting resistance of asphalt, reduces the temperature sensitivity of modified asphalt, and improves its storage stability. However, its effect on the low-temperature performance of asphalt is relatively minimal. Microscopic experimental results reveal the formation of a stable structure at the interface between GR and SBR in the composite modified asphalt. Furthermore, the dispersed phase exhibits improved uniformity, which positively impacts the stability of the asphalt binder.

**Keywords:** road engineering; graphene; styrene-butadiene rubber; rheological properties; micro analysis; analytic hierarchy process



## 1. Introduction

Asphalt, a commonly used pavement material in road construction, is typically composed of four main components: aromatic fraction, asphaltene, gum, and saturated fraction [1]. As the main laying type of modern highways, asphalt pavement has been widely used in several countries due to its high elastic bottom noise [2]. It has the characteristics of good pavement bearing capacity, driving comfort, no vibration, and low noise. However, in recent years, with the increase of traffic volume, unmodified pure asphalt binder poses challenges in meeting the requirements of modern traffic development and, therefore, necessitates modification [3]. Researchers have been studying the use of polymer modified asphalt (PMA) to obtain better asphalt pavement performance for several years [4]. In order to prevent early damage of the pavement and enhance the high and low temperature performance and water stability of the pavement, adding an appropriate amount of modifier is an effective way to improve the performance of asphalt.

Rubber modifier is a common modifier that has a great effect on improving the low temperature performance of asphalt. Styrene-butadiene rubber (SBR), as an early rubber modifier, is widely used because of its excellent low temperature road performance.

However, the high temperature performance of modified asphalt with small SBR content is not as good as its low temperature performance, which greatly limits the promotion of SBR modified asphalt [5]. In addition, the large differences in structure and solubility parameters between SBR and asphalt lead to potential instability of the blends [6]. These factors also limit the application scenarios of styrene-butadiene rubber modified asphalt in road engineering.

In recent years, the application of nano-modification in asphalt binders has demonstrated remarkable potential in meeting the evolving demands of the pavement industry. However, the influence of emerging carbon-based nanomaterials on asphalt binders, mortars, or concretes is an area that requires further investigation and study [7]. With the ongoing advancements in nanotechnology, researchers are continuously exploring the application of nanomaterials in fundamental engineering practices [8]. Researchers have found that the incorporation of nanomaterials into asphalt can solve the problems existing in many polymer modified asphalts, because nanomaterials generally have a large specific surface area that can hinder the free movement of asphalt, and their surface pores can also absorb the light components of asphalt. Numerous studies have demonstrated that the addition of nanoparticles such as anoclays (NC) [9], nano silicates [10], graphene [11] and others to asphalt can result in increased hardness and have been extensively investigated in the literature [12]. These nanoparticles have shown the ability to modify the mechanical properties of asphalt, giving it remarkable mechanical properties that increase the bitumen stiffness. Nanomaterial can also effectively improve the compatibility between polymer and asphalt. Because of these unique characteristics, more and more nanoparticles are used to improve the performance of asphalt materials. Graphene (GR) is a two-dimensional zero band gap half metal material [13]. It is a planar film composed of carbon atoms with $sp^2$ hybrid orbitals forming a hexagonal honeycomb lattice. It is a kind of magic nanomaterial. No matter what material it is added to, it seems to have a positive impact on the properties of the material [14,15]. Graphene has excellent mechanical properties [16]. As a new type of nanomaterial, the application of graphene in pavement engineering has attracted increasing attention [17]. Some scholars have prepared a 3D graphene mesh by chemical vapor deposition using petroleum asphalt as raw material. It is proved that asphalt is one of the suitable carbon sources for 3D graphene [18]. Its incorporation can significantly improve the high temperature performance of asphalt and mixture [19]. Graphene can also effectively avoid the defects of commonly used conductive phase materials and improve the conductivity of materials [20]. It also has broad prospects in the future application of snow melting pavement [21]. Graphene will produce a more effective response in the asphalt binder, but its elastic recovery ability is not as good as other asphalt modifiers. In addition, in the composite modification, graphene can reduce the thermal sensitivity of other binders by increasing heat transfer [22].

The high-temperature characteristics of SBR modified asphalt are generally subpar, and achieving uniform particle dispersion in the asphalt is challenging. Consequently, ameliorating the limitations of SBR asphalt has become a prominent area of research. Currently, there is limited research on composite modified asphalt using graphene and SBR. Therefore, this study aims to modify asphalt using graphene and styrene butadiene rubber, investigate the properties of the modified asphalt, and analyze the modification mechanism. The specific research method of this study is listed in Figure 1.

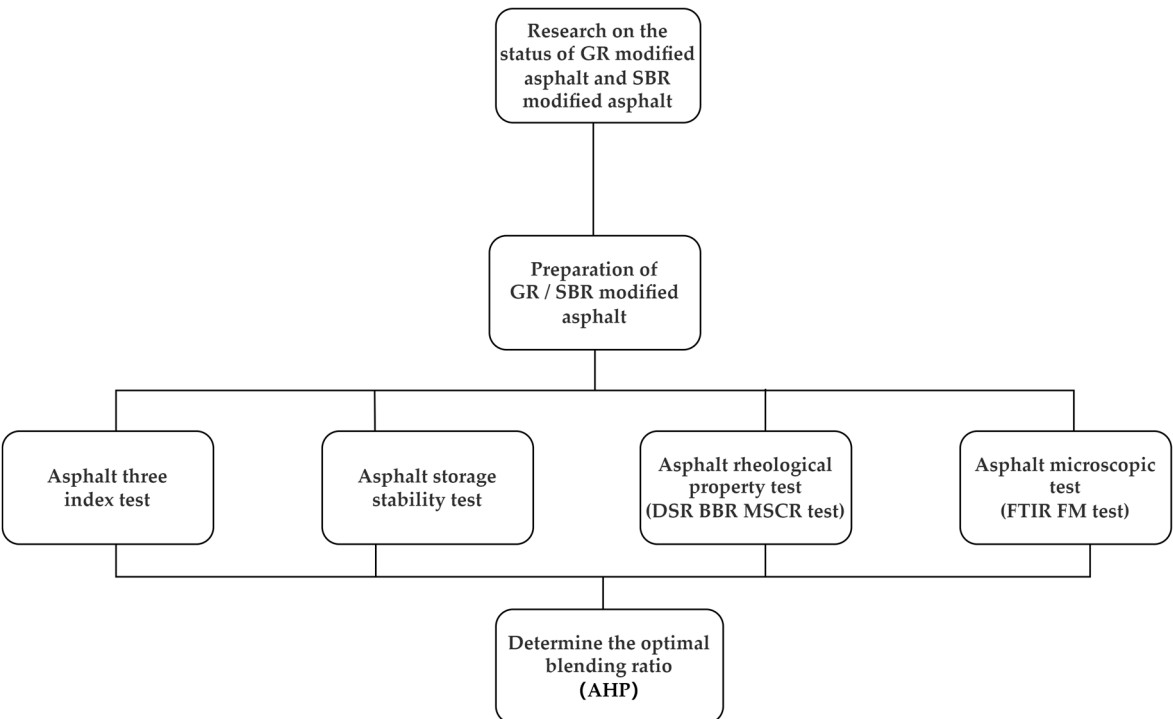

**Figure 1.** The design process of this study.

## 2. Materials and Methods

### 2.1. Materials

#### 2.1.1. Asphalt

Asphalt is a typical viscoelastic material [23]. It typically exhibits a semi-solid or solid state at room temperature, and its color is predominantly dark brown. In this study, the matrix asphalt utilized is SK-90 matrix asphalt. The assessment of asphalt performance is conducted in accordance with the relevant requirements and technical indicators specified in the asphalt test procedures. The performance indicators of the matrix asphalt are summarized in Table 1.

**Table 1.** Basic technical indicators of the 90-base asphalt used in this study.

| Test Index | Test Result | Test Requirement |
|---|---|---|
| Penetration (25 °C)/(0.1 mm) | 89.4 | 80–100 |
| Softening Point/°C | 45.3 | ≥45 |
| Ductility (5 °C/cm) | 11.5 | - |
| Brookfield Viscosity (60 °C/Pa·s) | 178.4 | ≥160 |

#### 2.1.2. Styrene-Butadiene Rubber (SBR)

SBR is a diblock copolymer. It consists of blocks of styrene units and butadiene units linked together in a linear arrangement. The SBR used in the experiment has a linear structure. This linear arrangement contributes to SBR's mechanical properties, such as its tensile strength and resilience. SBR modified asphalt has the characteristics of wear resistance, heat resistance, and aging resistance. It is prepared by adding SBR to asphalt and has excellent low temperature ductility and low temperature stress dissipation ability. Although SBR exhibits certain similarities to natural rubber in terms of its rubbery state, its physical mechanism performance, processing performance, and product performance are influenced by its unique composition and characteristics. SBR modifier has three main forms.: powder, emulsion, and rubber block. In this paper, SBR powder is used as modifier (Table 2), and the dosage is 4% for composite modification. Figure 2 depicts the SBR modifier.

**Table 2.** Technical indicators of the Styrene-butadiene rubber (SBR) used in the study.

| Test Index | Unit | Test Result |
|---|---|---|
| Morphology | - | white powder |
| Particle-size | Mesh | ≤40 |
| Filling oil content | % | 27.5 |
| Bound styrene content | % | 23.5 |
| Tensile strength | MPa | 28.5 |
| Tensile elongation | % | 410 |
| Molecular weight | thousand | 20–30 |

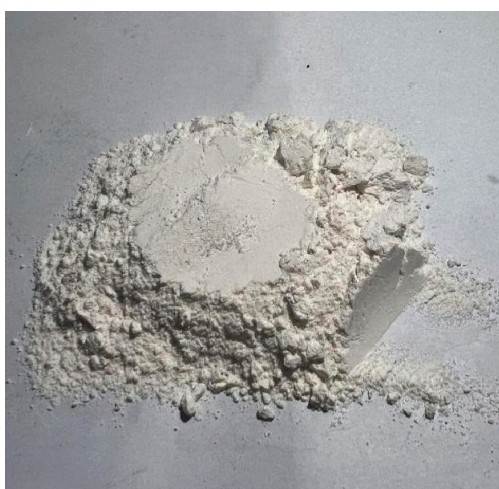

**Figure 2.** Styrene-butadiene rubber (SBR) modifier.

### 2.1.3. Graphene (GR)

Graphene (GR) is a two-dimensional material composed of a single layer of carbon atoms arranged in a hexagonal lattice. It is one of the strongest and most conductive materials known, with unique electronic, mechanical, and thermal properties. Graphene has been applied to the field of traffic engineering as a reinforcing phase. The addition of trace graphene to the polymer can change the microstructure of the asphalt binder, so that the asphalt has excellent mechanical, thermal, and adhesion properties [24]. The performance indicators of the graphene modifier used in this study are shown in Table 3. Figure 3 presents the graphene modifier.

**Table 3.** Technical indicators of the GR used in this study.

| Test Index | Unit | Test Result |
|---|---|---|
| Morphology | - | Black gray powder |
| Water content | % | ≤2 |
| Carbon content | % | ≥99 |
| Bulk density | g/mL | 0.01–0.02 |
| Grain size | $\mu m$ | 5–8 |

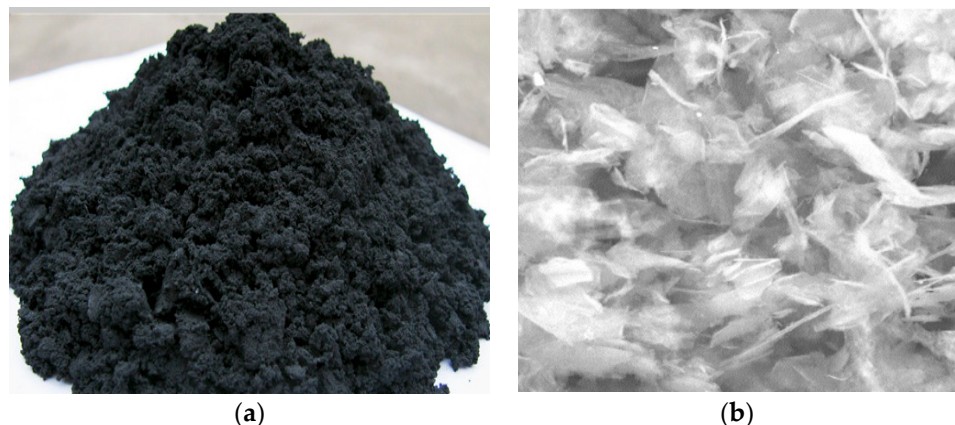

| (a) | (b) |

**Figure 3.** Graphene modifier. (**a**) Graphene powder, (**b**) Graphene under SEM.

*2.2. Preparation Methods*

Asphalt Modification Procedure

To ensure thorough mixing of the modifier and matrix asphalt, two methods were employed for preparing the modified asphalt: the melt blending method and the mechanical blending method. In the melt blending method, the matrix asphalt was heated to a temperature of 155 °C until it reached a fully flowing state. It was then transferred to an electric hot plate set at a temperature of 160 °C. The SBR modifier was added, and then the stirring commenced. Then, 0.02%, 0.04%, 0.06%, 0.08%, and 0.1% graphene were added. The electric heating plate was heated to 160 °C and stirred at 500 r/min for 30 min and then put into the mixer. The rotor speed was set to 5000 r/min, and the high-speed shear at 160 °C for 60 min. Finally, the graphene and styrene butadiene rubber composite modified asphalt are obtained by stirring and swelling for 30 min. The specific preparation process is as Figure 4.

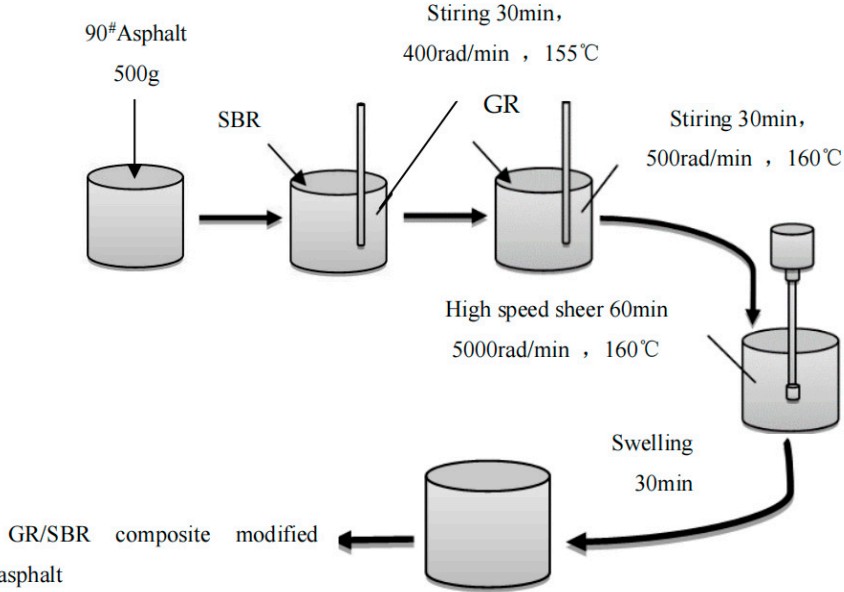

**Figure 4.** Composite modified asphalt preparation flow chart.

*2.3. Test Methods*

2.3.1. Routine Performance Test of Asphalt

The softening point (R and B), penetration at 25 °C, and ductility at 5 °C of the GR/SBR modified asphalt were assessed in accordance with the testing protocols specified in the

Standard Test Methods of Bitumen and Bituminous Mixtures for Highway Engineering (JTGE20-2011) [25].

### 2.3.2. Storage Stability Analysis of Asphalt

The stability of the polymer modified asphalt in the process of use is also one of the key issues that researchers have been paying attention to. Therefore, in order to study whether the modified asphalt has good storage stability, the segregation test (storage stability test) was carried out on the modified asphalt with different blending ratios to determine whether the modified asphalt will be segregated during use and storage. First, the molten asphalt was placed in an aluminum tube with a diameter of 25 mm and a height of 140 mm, and then sealed and placed vertically in an oven at 160 °C for 48 h. Finally, the storage stability of the modified asphalt was evaluated by measuring the difference of softening point between the top and bottom samples of aluminum tube. If the softening point difference between the top and bottom is less than 2.5 °C, the sample is considered to have good storage stability and compatibility.

### 2.3.3. Dynamic Shear Rheometer Test

Asphalt is a viscoelastic material with rheological properties, which means that the viscoelasticity and fluidity of asphalt are closely related to temperature. At high temperatures, the viscous components of asphalt increase and the fluidity of asphalt becomes better, which leads to the decrease of rutting resistance of asphalt concrete. The temperature scanning experiment was conducted using a dynamic shear rheometer (Anton Paar MCR 302 DSR instrument, Austria). This instrument allows for precise measurement of the asphalt's response to dynamic shear forces at various temperatures. The test is carried out according to the requirements of AASHTO TP70-11 [26]. Continuous sinusoidal alternating load is applied, and strain control is adopted. The diameter of the test metal plate is 25 mm, and the thickness of the asphalt sample is 1.0 mm. The test temperature is 30~85 °C, the strain level is 1%, and the angular frequency is 10 rad/s.

### 2.3.4. Bending Beam Creep Test at Low Temperatures

Bending creep tests of modified asphalt were conducted using the TE-BBR-F (CAN-NON, Melville, NY, USA) instrument at temperatures of −12 °C, −18 °C, and −24 °C. The loading process was 240 s. The low temperature crack resistance of the modified asphalt was evaluated according to the stiffness modulus and creep rate recorded in the 60th second [27].

### 2.3.5. Infrared Spectrum Test

To prepare a 10% solution, the modified asphalt sample was dissolved in dichloromethane. Fourier transform infrared spectroscopy (FTIR) was used to study the effect of the graphene modifier on the chemical composition and functional groups of the modified asphalt under different dosages. Its resolution is 4 cm$^{-1}$, the number of scans is 32, and the wavenumber test range is 400~4000 cm$^{-1}$ [28].

### 2.3.6. Fluorescence Microscope

The analysis of polymer modifier dispersion in asphalt was conducted using a high-resolution fluorescence microscope (AxiolmagerA2, Zeiss, Germany). In the fluorescence microscopy test, the two-photon laser confocal super-resolution microscope was used; the magnification was set to 10 × 10.

## 3. Results and Discussion

### 3.1. Conventional Performance Test

#### 3.1.1. Analysis of Basic Physical Properties of GR/SBR Modified Asphalt

The three major indicators of asphalt were tested in accordance with the relevant requirements and test methods. At the same time, the penetration (25 °C) after short-term

aging was obtained by the experiment, and the residual penetration ratio was calculated. The specific results are shown in the figure.

From the test results shown in Figure 5, the penetration of SBR modified asphalt shows a downward trend with the increase of graphene incorporation, and the softening point continues to increase. When the GR content is 0.1%, the softening point is the highest (61.4 °C). This shows that the incorporation of graphene improves the high temperature performance of the modified asphalt. The incorporation of SBR reduces the penetration of asphalt, and the softening point and ductility are improved when compared with matrix asphalt, indicating that the incorporation of SBR increases the consistency and flexibility of asphalt and enhances the high temperature deformation resistance and low temperature ductility. After the composite modification, the penetration continues to decrease, the softening point further increases, and the ductility is significantly higher than that of the matrix asphalt. Therefore, the incorporation of graphene enhances and improves the high temperature performance of asphalt. This is because, in the process of composite modification, graphene and swollen rubber powder are entangled with each other, which enhances the internal network structure of modified asphalt, so that its deformation resistance is improved. However, the incorporation of graphene will adsorb the light components in the asphalt, making its low temperature ductility decrease. The residual penetration ratio of asphalt is the ratio of residual penetration to initial penetration after long-term aging under certain conditions. Short-term aging of asphalt means that asphalt is affected by external conditions such as high temperature and oxidation in a short time, resulting in changes in its physical properties. This aging process causes the asphalt molecular chains to fracture and crosslink, resulting in increased density and hardness. The experimental results show that the residual penetration ratio of modified asphalt after short-term aging first increases and then decreases with the increase of the content of graphene. When the content of graphene is 0.04%, the residual penetration ratio is 81.2%, which shows that the incorporation of graphene can effectively improve the anti-aging performance of asphalt.

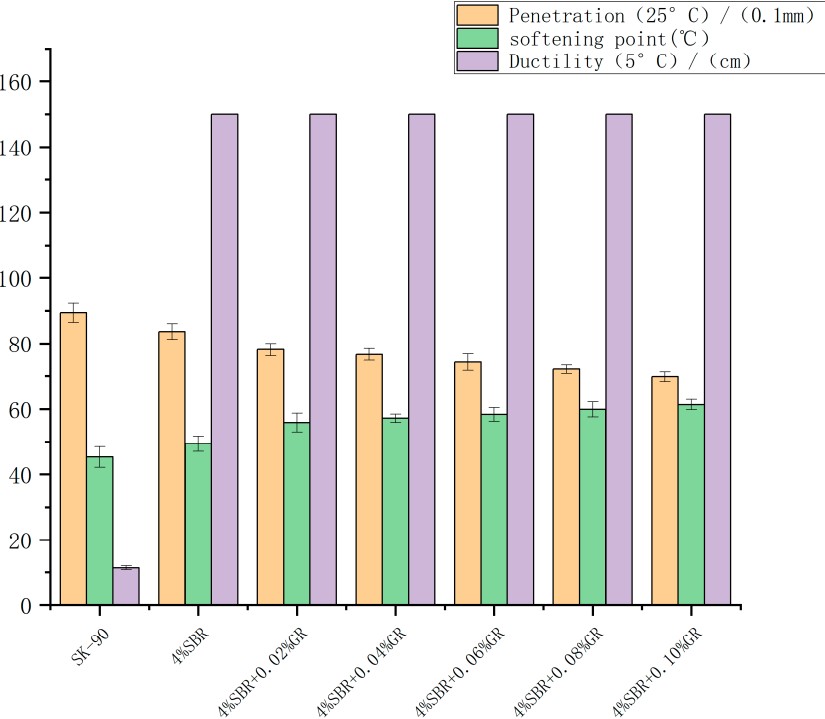

**Figure 5.** Effect of GR content on three modified asphalt indexes.

### 3.1.2. Storage Stability Analysis of GR/SBR Modified Asphalt

In general, factors such as the weight difference between polymer and pitch molecules can make them incompatible at the molecular level. This will cause the polymer and asphalt in the modified asphalt to exist in two different phases. If the two phases are not well compatible at the phase interface, it will lead to a poor connection between the polymer and the asphalt, affecting the modification effect. In order to meet the requirements of practical application and storage, and to ensure a good modification effect, we carried out the test according to the softening point test procedure and calculated the softening point difference of the modified asphalt after standing for 48 h at different parts (top and bottom) of the asphalt sample tube. If the difference does not exceed 2.5 °C, it shows that it has good storage stability.

It is evident from Table 4 and Figure 6 that the addition of only 4% SBR results in the largest difference between the upper and lower softening points of asphalt (2.9 °C). However, upon introducing the graphene modifier, the softening point difference of the modified asphalt becomes smaller when compared to the SBR modified asphalt. This indicates that the inclusion of graphene reduces the disparity between the upper and lower softening points of asphalt, leading to an improvement in the segregation phenomenon.

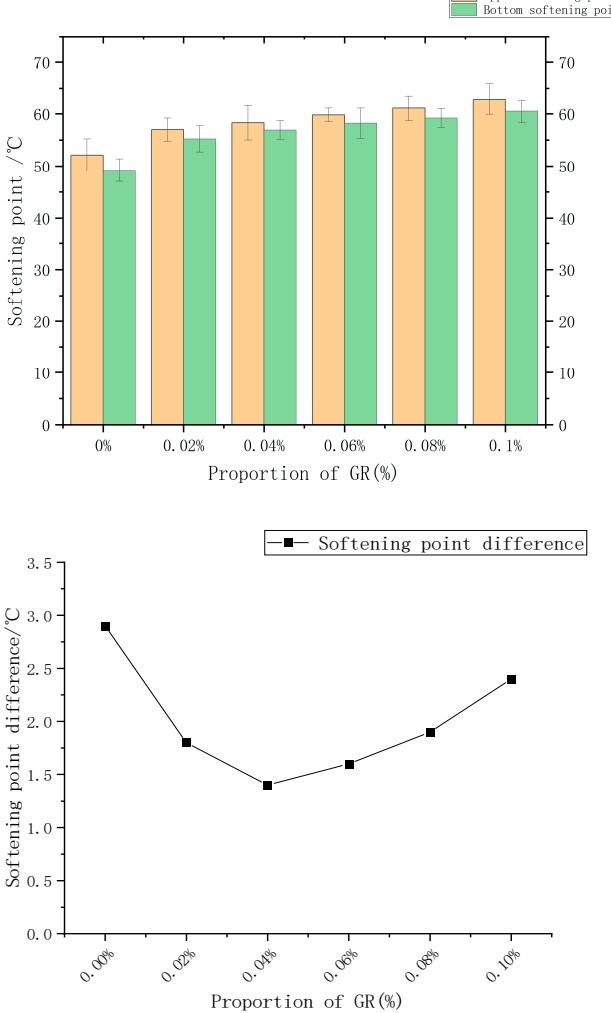

**Figure 6.** Effect of graphene content on storage stability of modified asphalt.

**Table 4.** The softening point of each part of asphalt after segregation experiment.

| Types of Asphalt | Upper Softening Point | Bottom Softening Point | Softening Point Difference |
|---|---|---|---|
| 4% SBR | 52.1 | 49.2 | 2.9 |
| 4% SBR + 0.02% GR | 57.0 | 55.2 | 1.8 |
| 4% SBR + 0.04% GR | 58.3 | 56.9 | 1.4 |
| 4% SBR + 0.06% GR | 59.8 | 58.2 | 1.6 |
| 4% SBR + 0.08% GR | 61.1 | 59.2 | 1.9 |
| 4% SBR + 0.1% GR | 63.2 | 60.8 | 2.4 |

Furthermore, when the graphene content reaches 0.04%, the softening point difference becomes the smallest, measuring 1.4 °C. This can be attributed to the incorporation of graphene, which strengthens the phase interface, forms cross-links with SBR within the asphalt, promotes a more uniform dispersion of the phase, and establishes a stable structure. Consequently, this enhances the control of asphalt molecules within the dispersed structure and improves the compatibility between the polymer and asphalt.

### 3.2. Rheological Properties of GR/SBR Modified Asphalt

3.2.1. Temperature Scanning Test and Analysis

The composite shear modulus, denoted as G*, is a measure of the sample's ability to withstand deformation under repeated shear stress. Examining Table 5, it is evident that within the experimental temperature range of 58 °C to 76 °C, the complex shear modulus G* of each asphalt decreases as the temperature rises. Figure 7 indicates a reduction in the asphalt's ability to resist load-induced deformation with increasing temperature. This decrease in performance can be attributed to the weakening of intermolecular forces and crosslinking relationships among asphalt molecules as temperature rises. Consequently, the elasticity of the asphalt decreases while the viscous component increases, leading to a decline in the asphalt's resistance to deformation. Notably, at 64 °C and with a graphene content of 0.1%, the composite modified asphalt demonstrates a G* value that is 138.2% higher when compared to the SBR modified asphalt. This indicates that the addition of graphene alters the viscoelastic composition of the asphalt. The composite modified asphalt exhibits a greater proportion of elastic components at the same temperature, thereby enhancing its ability to withstand deformation at high temperatures.

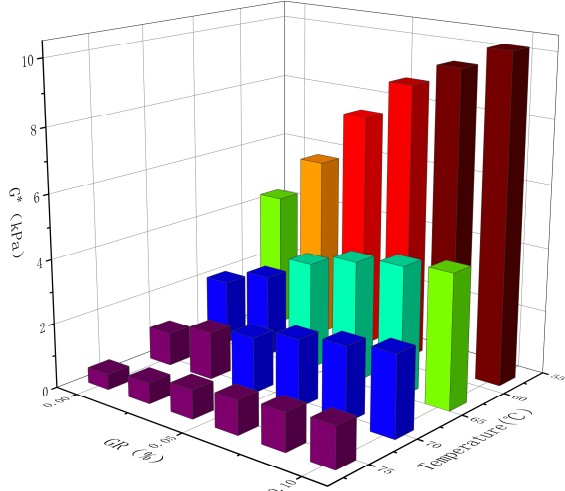

**Figure 7.** Effect of temperature and GR content on modified asphalt G*.

**Table 5.** Influence of temperature and GR content on G* (kPa) of modified asphalt.

| GR Content (%) | 58 °C | 64 °C | 70 °C | 76 °C |
|---|---|---|---|---|
| 0 | 4.281 | 2.021 | 1.017 | 0.473 |
| 0.02 | 5.792 | 2.565 | 1.453 | 0.664 |
| 0.04 | 7.535 | 3.362 | 1.716 | 0.912 |
| 0.06 | 8.771 | 3.794 | 2.087 | 1.068 |
| 0.08 | 9.489 | 4.027 | 2.319 | 1.187 |
| 0.10 | 10.198 | 4.196 | 2.532 | 1.265 |

Table 6 and Figure 8 illustrate that the modified asphalt containing 0.10% graphene exhibits the highest rutting factor across the entire range of experimental temperatures, reaching a maximum value of 10.312 kPa at 58 °C. The higher the rutting factor, the better the high-temperature stability [29]. The rutting factor serves as an indicator of the asphalt's resistance to rutting, with higher values indicating greater resistance to deformation at the same temperature. In accordance with ASTM D7643-10. specification requirements, the experimental results indicate a significant increase in the rutting factor of the composite modified asphalt with rising temperatures. This suggests improved resistance to permanent deformation under high-temperature conditions. Additionally, the rutting factor initially decreases sharply and then gradually levels off as the temperature increases, implying a decrease in the temperature sensitivity of the asphalt's rutting factor with rising temperatures. Moreover, the incorporation of graphene leads to an increase in the asphalt's rutting factor, indicating that graphene effectively enhances the asphalt's resistance to high-temperature rutting. Furthermore, a higher graphene content yields better high-temperature performance of the modified asphalt. This can be attributed to the increased cohesion of the asphalt binder resulting from the inclusion of graphene, thereby enhancing the high-temperature resistance of the modified asphalt.

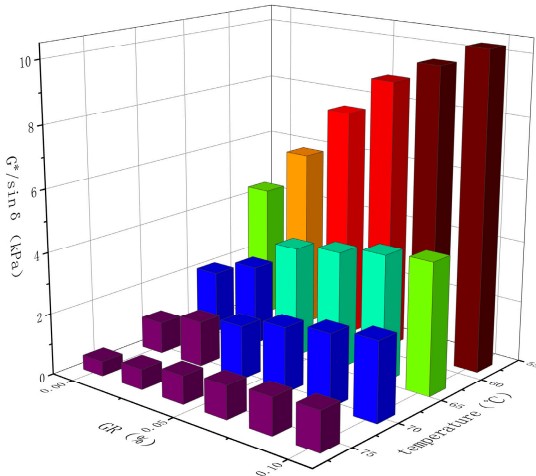

**Figure 8.** Effect of temperature and GR content on modified the G*/sin δ of modified asphalt.

**Table 6.** Influence of temperature and GR content on G*/sin δ (kPa) of modified asphalt.

| GR Content (%) | 58 °C | 64 °C | 70 °C | 76 °C |
|---|---|---|---|---|
| 0 | 4.369 | 2.062 | 1.028 | 0.476 |
| 0.02 | 5.908 | 2.631 | 1.466 | 0.671 |
| 0.04 | 7.617 | 3.643 | 1.748 | 0.925 |
| 0.06 | 8.872 | 3.872 | 2.131 | 1.087 |
| 0.08 | 9.591 | 4.154 | 2.364 | 1.205 |
| 0.10 | 10.312 | 4.301 | 2.583 | 1.289 |

### 3.2.2. Multiple Stress Creep Recovery Test Analysis

The Multiple Stress Creep Recovery (MSCR) test is a test method developed to solve the problem that the original evaluation system is not applicable to modified asphalt [30]. The results indicate that the MSCR test is capable of effectively capturing the nonlinear viscoelastic behavior of polymer modified asphalt at high temperatures. To comprehensively assess the high temperature performance of graphene and styrene butadiene rubber composite modified asphalt, the MSCR test is employed in conjunction with the Dynamic Shear Rheometer (DSR). By subjecting the asphalt sample to applied stress, the MSCR test records both delayed elastic recovery deformation and unrecoverable deformation. Upon removal of the stress, certain deformation is recovered, while the unrecoverable deformation accumulates for the subsequent cyclic load. This cyclic loading and unloading process simulates the stress and deformation experienced by the pavement during actual usage. The repeated stress creep test effectively demonstrates the modified asphalt's capacity for elastic recovery, closely simulates its characteristics of minimal cumulative deformation and high deformation resistance, and exhibits a strong correlation with rut depth. The average strain recovery rate (*R*) and unrecoverable creep compliance (*Jnr*) are commonly used to evaluate the test results. The formulas of *R* and *Jnr* are shown as following Formulas (1) and (2):

$$R = 0.1 \sum_{i=1}^{10} \frac{\sigma_{ip} - \sigma_{inr}}{\sigma_{ip} - \sigma_{io}} \tag{1}$$

$$J_{nr} = 0.1 \sum_{i=1}^{10} \frac{\sigma_{inr} - \sigma_{io}}{\tau} \tag{2}$$

In the formula: $\sigma_p$ is the peak strain of each period; $\sigma_o$ is the initial strain of each period; $\sigma_{nr}$ is the residual strain of each period; and is $\tau$ for loading stress.

It can be seen from the Figures 9 and 10 that under two different stress levels, No. 90 matrix asphalt has the highest *Jnr* value and the lowest *R* value, indicating that matrix asphalt is easier to accumulate deformation and has poor deformation recovery ability.

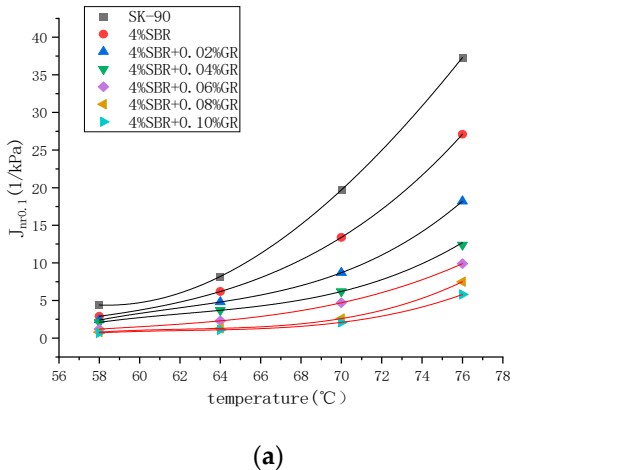
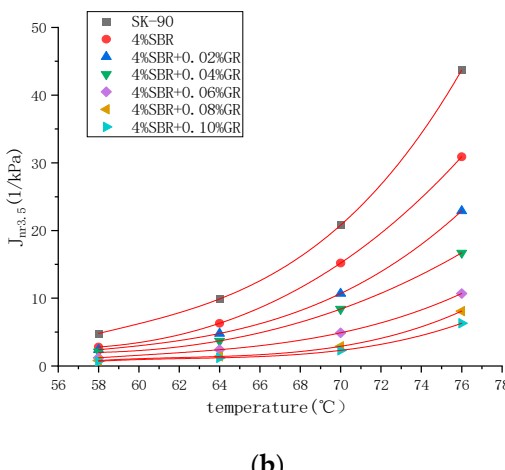

(**a**) (**b**)

**Figure 9.** Changes in the unrecoverable creep compliance of asphalt with temperature under different stress levels. (**a**) 0.1 kPa, (**b**) 3.2 kPa.

Based on the experimental results, it is evident that at higher stress levels (3.2 kPa), the R value of each asphalt decreases, while the Jnr value increases. This indicates that under the influence of stress, higher stress levels can negatively impact the anti-deformation ability of the asphalt, resulting in a weakened ability to recover from deformation. Additionally, it suggests that the asphalt's resistance to deformation is also compromised. Furthermore, as the temperature increases, the R value of each asphalt decreases and the Jnr value increases.

This indicates that elevated temperatures, combined with high stress levels, contribute to a reduction in the asphalt's ability to recover from deformation and its resistance to deformation.

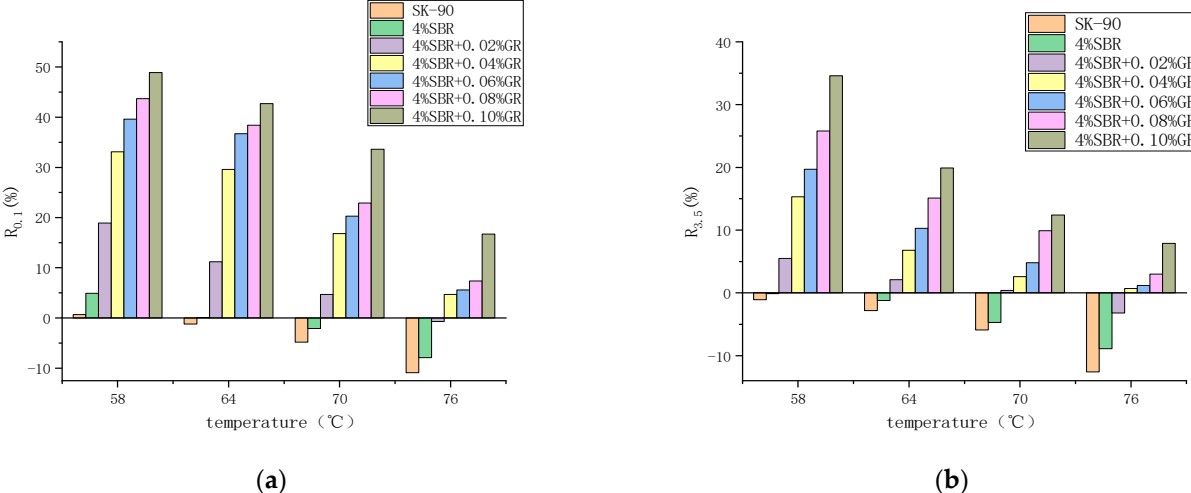

**Figure 10.** Changes in the average strain recovery rate of asphalt with temperature under different stress levels. (**a**) 0.1 kPa, (**b**) 3.2 kPa.

We can also see that as the experimental temperature increases, the Jnr of each asphalt increases nonlinearly. As the temperature increases, the slope of the curve increases significantly. However, when compared with the 90# matrix asphalt, the incorporation of the modifier makes the second derivative value of the curve correspond to the Jnr value decrease, that is, the slope of the Jnr curve decreases with the increase of temperature. With the increase of graphene content, the rate of increase of the Jnr curve with temperature also decreases, indicating that the incorporation of graphene can reduce the temperature sensitivity of asphalt. This is because in the process of composite modification, graphene and swollen SBR are entangled with each other. The internal network structure of the modified asphalt is enhanced, thereby blocking the temperature and reducing the temperature sensitivity of the asphalt. At the same time, when the content of graphene is 0.02% and 0.04%, the viscoelastic properties of asphalt are obviously improved, and with the gradual increase of the content, the marginal effect is produced, and the improvement of viscoelastic properties is reduced.

### 3.3. Low-Temperature Cracking Performance of GR/SBR Modified Asphalt

The BBR (Bending Beam Rheometer) test is used to assess the impact of graphene content on the low-temperature creep performance of asphalt, providing insights into the asphalt's performance under low-temperature conditions. The test involved evaluating the creep properties of graphene-modified asphalt at temperatures of −12 °C, −18 °C, and −24 °C. The stiffness modulus (S) and creep rate (m) at 60 s were utilized as evaluation criteria. The creep stiffness (S) and creep rate (m) obtained from the bending beam rheometer experiment offer insights into the asphalt's resistance to low-temperature cracking. The smaller the value of S and the larger the value of m, the better the asphalt low-temperature performance [31]. Generally, larger S values indicate greater asphalt hardness and an increased susceptibility to cracking at low temperatures. On the other hand, larger m values indicate a stronger stress relaxation ability of the asphalt, slower stress accumulation, and improved low-temperature performance. The changes in S and m values of the graphene-modified asphalt with varying graphene content are depicted in Figures 11 and 12, respectively.

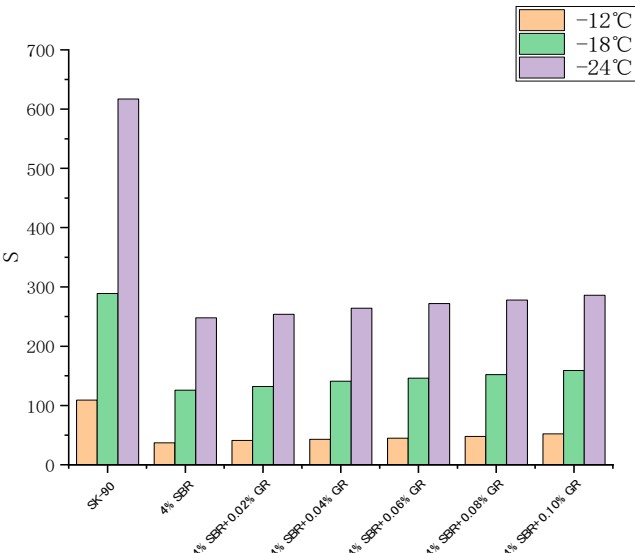

**Figure 11.** Effect of GR content on the stiffness modulus of modified asphalt.

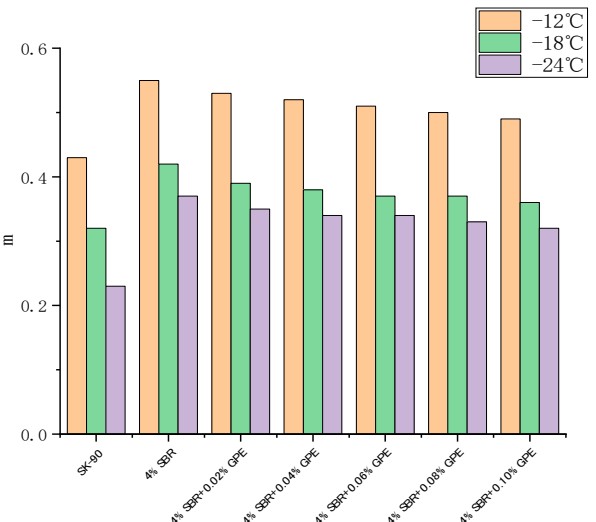

**Figure 12.** Effect of GR content on the creep rate of modified asphalt.

The experimental results demonstrate that the creep rate (m) of 90# asphalt, SBR modified asphalt, graphene, and SBR/GR modified asphalt decreases as the temperature decreases. This indicates that the stress relaxation ability of the asphalt material diminishes with decreasing temperature. Conversely, the stiffness modulus (S) of each asphalt increases as the temperature decreases. This suggests that the asphalt material becomes more brittle with lower temperatures.

In accordance with ASTM D7643-10 [32] specification requirements, the stiffness modulus S should be less than 300 MPa and the creep rate m should not be less than 0.3 when the test time is 60 s. The diagram reveals that the S value of the matrix asphalt exceeds 300 MPa, while the m value falls below 0.3 at −24 °C, thereby failing to meet the established standard. Conversely, at −18 °C, the S value complies with the standard, whereas the m value fails to meet the specified requirements. Meanwhile, SBR modified asphalt and graphene and SBR composite modified asphalt meet the requirements.

It can be seen from the experimental results that at the same temperature, the stiffness modulus S of modified asphalt with 4% SBR is significantly lower than that of matrix asphalt, and the creep rate m is significantly improved, indicating that the incorporation of SBR improves the low temperature crack resistance of asphalt.

As there are some differences and limitations in evaluating the low temperature performance of rubber asphalt with S and m indexes at different temperatures, it is necessary to find an index that can consider both S and m in a wide temperature range. Therefore, in order to consider the low temperature cracking resistance and stress relaxation ability of modified asphalt at the same time, the k index is introduced. The *k* index has a good correlation with the mixture. The test is fast and convenient, and the test accuracy is high. It can accurately distinguish the low temperature performance of matrix asphalt and rubber asphalt and different rubber asphalt [33]. The formula for calculating the *k*-value is as following Formula (3):

$$k = \frac{S}{m} \tag{3}$$

S is the stiffness modulus and m is the creep rate of modified asphalt.

The relationship between the stiffness modulus of modified asphalt and its creep rate at the same temperature is inverse. This means that a lower stiffness modulus is indicative of better low temperature performance in the asphalt. Consequently, a smaller value of *k* reflects enhanced low temperature crack resistance and improved relaxation ability in the asphalt. The calculation results are shown in Figures 3–7:

Figure 13 clearly shows that the addition of SBR significantly reduces the *k*-index of asphalt, indicating a significant improvement in the asphalt's low-temperature performance. However, with the incorporation of graphene, the *k*-index of asphalt gradually increases, suggesting a decrease in its low-temperature performance. This can be attributed to the adsorption of light components in the asphalt due to the presence of graphene, leading to a decrease in low-temperature ductility. However, the decrease in the k-index is not substantial.

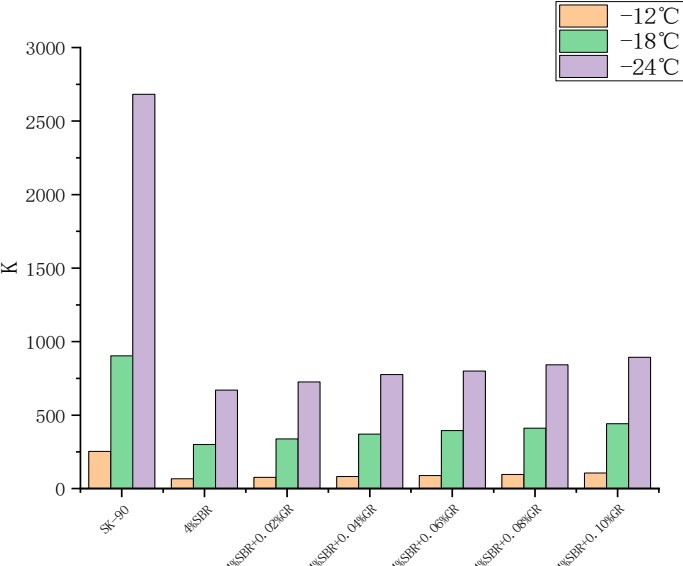

**Figure 13.** Influence of GR content on the k index of asphalt.

For instance, at −18 °C, the k-value of the asphalt reaches its peak when the graphene (GR) content is 0.1%, with a k-value of 441.7. This value is considerably lower than the k-value of the base asphalt, which stands at 903.1 at the same temperature. This indicates that although the low-temperature performance of the graphene-modified asphalt experiences a certain decline, it still exhibits significantly stronger low-temperature performance compared to the base asphalt.

*3.4. Study on Modification Mechanism of Graphene*

3.4.1. Fourier Infrared Spectrum

Infrared spectroscopy is one of the most widely used methods to study the chemical structure of polymers [34]; it is also widely used to study the chemical structure of asphalt. [35] The Fourier infrared spectrum obtained by FTIR test is shown in the figure. Through infrared spectrum analysis, the influence of additives on the structure of asphalt and the interaction and reaction between different components were explored, and their functional groups and chemical bonds were analyzed and identified. The band position and intensity in the infrared spectrum image can reflect the vibration characteristics of different chemical bonds, which provides a powerful tool for analyzing the composition and structure of asphalt. In the infrared spectrum, distinct absorption peaks can be observed. Through the judgment of each absorption peak, the modification mechanism of asphalt modified by modifier is quantitatively analyzed.

In Figure 14, different types of vibration occur when the atoms in the molecule absorb infrared radiation at a specific wavelength. Among them, it mainly includes two types of vibration: tensile vibration and bending vibration. Tensile vibration refers to the relative motion of two atoms in a molecule along the direction of a bond, which is the same as the bond axis. The bending vibration refers to the bending motion of two atoms in the molecule around the bond axis. Therefore, bending vibration usually does not lead to significant changes in bond length.

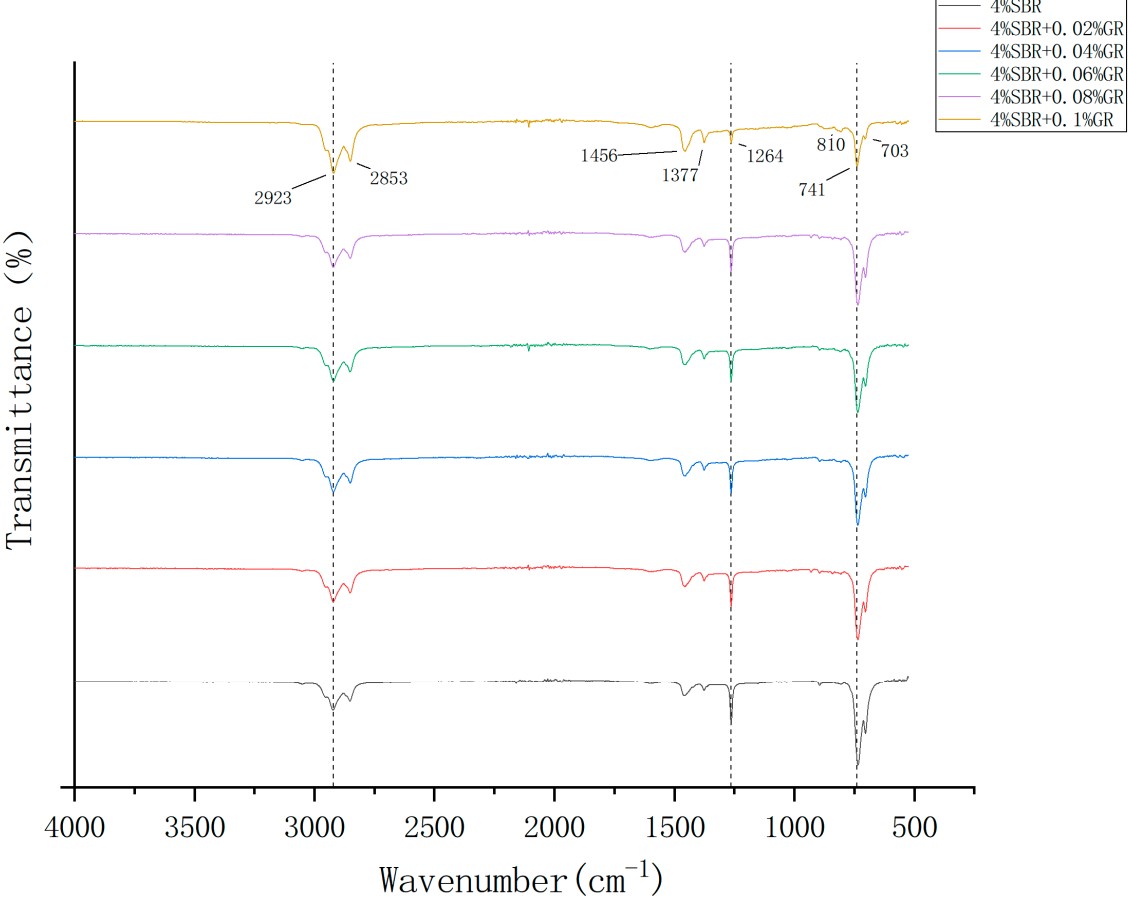

**Figure 14.** Infrared spectra of modified asphalt.

Analysis of the infrared spectrum reveals that the characteristic peaks of graphene and styrene-butadiene rubber (SBR) composite modified asphalt and SBR modified asphalt are essentially identical. The only difference lies in the intensity of the absorption peaks, and no additional absorption peak is observed within the entire functional group region of

the modified asphalt. This indicates that the modification of asphalt using graphene is a physical modification process that does not involve the addition of other modifiers and does not alter the chemical composition of the asphalt.

The cis-methyl group refers to the bond angle between the carbon atoms of the two adjacent methyl groups (about 60 degrees), and the trans-methyl group refers to the bond angle between the carbon atoms of the two adjacent methyl groups (about 180 degrees). In the infrared spectrum, the C-H bond in the cis-methyl group usually shows two different absorption peaks, a symmetric stretching vibration peak and an asymmetric stretching vibration peak; the C-H bond in the trans-methyl group usually exhibits a separate absorption peak, and the symmetric stretching vibration peak and the asymmetric stretching vibration peak are combined to form a wide absorption peak. By analyzing Figure 14, we can see that the C-H bond in the wave number of 2800–3000 cm$^{-1}$ has two different absorption peaks, and the characteristic peaks are obvious. Therefore, the methyl group here is a cis-methyl group. The stretching vibration of C-H here is the C-H bond swing vibration in the butadiene molecule, which also shows that the compatibility of styrene-butadiene rubber and asphalt is good, and the main modification method is physical modification.

We can also see from Figure 14 And Table 7 that there is a more obvious absorption peak in the wave number of 700–850 cm$^{-1}$, which is due to the C-H bending vibration in the aromatic compounds in the modified asphalt. At the same time, we also note that with the increase of graphene content, the corresponding transmittance of the material in the infrared spectrum decreases here. This is because the bending vibration frequency of the C-H group in graphene is very close to the infrared absorption peak. Its incorporation absorbs the incident infrared radiation energy, resulting in a decrease in transmittance. With the increase of graphene content, more C-H groups are involved in this bending vibration, and this phenomenon becomes more significant, which also shows that the modification method of graphene for asphalt is mainly physical modification.

**Table 7.** The main characteristic peaks and corresponding wave numbers in the infrared spectrum.

| Wave Number (cm$^{-1}$) | Characteristic Peak |
| --- | --- |
| 2923 | Asymmetric stretching vibration of methyl group C-H |
| 2853 | C-H symmetric stretching vibration of methyl group |
| 2107 | The triple bond stretching vibration of cyano (-CN) |
| 1456 | Bending vibration of C-H in hydrocarbons |
| 1377 | C-H Bending Vibration in Aromatic Compounds |
| 1264 | The bending vibration of carboxyl (C=O) |
| 810 | |
| 741 | C-H Bending Vibration in Aromatic Compounds |
| 703 | |

### 3.4.2. Fluorescence Microscope Images and Compatibility Analysis

Figure 15 shows images of polymer modified asphalt, including two different phases: asphalt phase and polymer phase. Asphalt and polymer emit different colors under different light sources under a fluorescence microscope, so the dispersion and particle size of the polymer in the asphalt can be observed and analyzed by a fluorescence microscope. Figure 15 corresponds to the 90# asphalt, SBR modified asphalt, and SBR/GR modified asphalt, respectively. The relatively bright part is the color emitted by polymer excitation, while the asphalt is the darker part in the picture. It can be seen from the figure that the size and dispersion of SBR particles in the SBR modified asphalt and the SBR/GR composite modified asphalt are roughly the same, and there is a good interface phase between SBR and asphalt; this indicates that SBR and asphalt have good compatibility. At the same time, the addition of GR is uniform for the dispersed phase of SBR in asphalt, which is in SBR/GR composite modified asphalt. Because GR molecules are dispersed in asphalt

and SBR particles in small size, and some GR and SBR interface contact to form a stable structure, these GR molecules play a good role in heat insulation and protection, thus reducing the impact of high temperature and aging on asphalt and polymer. In summary, the composite modified asphalt formed by the incorporation of an appropriate amount of GR and SBR will form a stable structure with the asphalt. It is conducive to improving the performance of the asphalt.

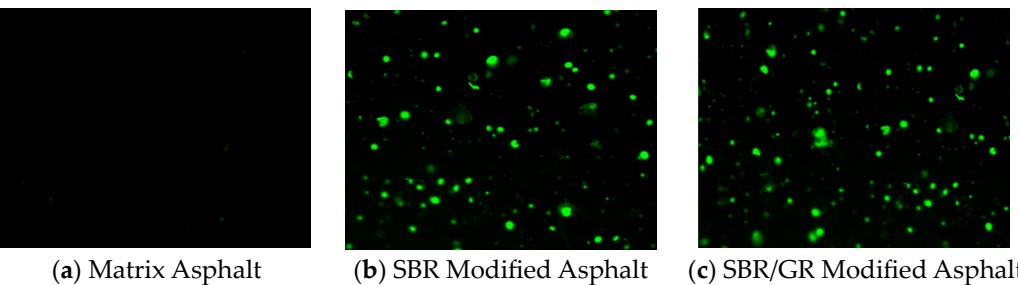

| (**a**) Matrix Asphalt | (**b**) SBR Modified Asphalt | (**c**) SBR/GR Modified Asphalt |

**Figure 15.** Fluorescence microscope (10 × 10) imagery.

### 3.5. Optimization of GR/SBR Composite Asphalt

The comprehensive evaluation methods commonly used in the study include the Delphi method, the SAW method, AHP, and the fuzzy comprehensive evaluation method [36]. Analytic hierarchy process (AHP) is a systematic method for multi-criteria decision analysis. It was first proposed by American scholar Thomas L. Saaty in the 1970s. The purpose of this method is to cluster similar variables and then analyze each cluster to determine the structure and relationship between variables. This method first groups the variables according to the similarity of the variables, and then uses the combination of statistics and visualization tools to explore the relationship between the variables in each cluster. Due to the systematic and hierarchical nature of the analytic hierarchy process, the establishment of the analytic hierarchy process model can make the complex problems organized, and the use of quantitative methods to reflect the subjective judgment on the importance of each factor, in order to avoid the error caused by subjective judgment. This chapter uses the analytic hierarchy process to optimize the graphene content in graphene and styrene-butadiene rubber composite modified asphalt.

3.5.1. Evaluation Index Optimization of Modified Asphalt

Through the summary of previous scholars' research, combined with the characteristics of large temperature difference and low winter temperature in Northeast China, the k index at $-18\,°C$ was employed to assess the low-temperature performance of composite modified asphalt. The rutting factor $G^*/\sin\delta$ at 64 °C was utilized to evaluate the high-temperature characteristics of the asphalt. The temperature sensitivity of the modified asphalt was evaluated using the penetration index (PI) value. The anti-aging performance of the modified asphalt was assessed through the residual penetration after short-term aging. The storage stability of modified asphalt was evaluated by the softening point difference of asphalt after segregation experiment.

The calculation of penetration index is as follows in Formula (4):

$$PI = \frac{20 - 500A_{\lg Pen}}{1 + 50A_{\lg Pen}} \tag{4}$$

In the formula: $A_{\lg Pen}$ is the slope of the penetration regression line at different temperatures. The greater the value of penetration index *PI*, the lower the sensitivity of asphalt to temperature.

### 3.5.2. Modified Asphalt Evaluation Index Scoring Criteria

On the premise of meeting various technical requirements, the above-mentioned preferred evaluation indexes are divided into four grades: excellent, good, medium, and passing. The average scores corresponding to each grade are 9, 8, 7, and 6, respectively. When an index does not meet the requirements of the specification, the asphalt cannot be used in the project, is deleted, counted as 0 points, and does not participate in the comprehensive performance evaluation. The number of each asphalt is shown in Table 8; the adjacent grade score is determined according to Table 9; the modified asphalt evaluation index data is shown in Table 10; and the data score value of each group of modified asphalt is shown in Table 11.

**Table 8.** GR content of modified asphalt.

| Grade | GR (%) |
|---|---|
| A | 0 |
| B | 0.02 |
| C | 0.04 |
| D | 0.06 |
| E | 0.08 |
| F | 0.1 |

**Table 9.** Determination of adjacent grade scores.

| Mark | 0 | 6 | 7 | 8 | 9 |
|---|---|---|---|---|---|
| G*/sinδ (64 °C)/KPa | <2 | 2–2.5 | 2.5–3 | 3–4 | >4 |
| k (MPa) | >450 | 400–450 | 350–400 | 300–350 | <300 |
| Penetration index *PI* | <−0.9 | −0.9–0.7 | −0.7–0.5 | −0.5–0.3 | >−0.3 |
| Residual penetration ratio (25 °C)/% | <65 | 65–70 | 70–75 | 75–80 | >80 |
| The softening point difference of asphalt after segregation experiment (25 °C) | >3 | 2.5–3.0 | 2.0–2.5 | 1.5–2.0 | <1.5 |

**Table 10.** Summary of evaluation indexes of each modified bitumen group.

| Asphalt Type | A | B | C | D | E | F |
|---|---|---|---|---|---|---|
| G*/sinδ (64 °C)/KPa | 2.062 | 2.631 | 3.643 | 3.872 | 4.154 | 4.301 |
| k (MPa) | 300.06 | 338.46 | 371.05 | 394.59 | 410.81 | 441.67 |
| Penetration index PI | −0.73 | −0.41 | −0.29 | −0.37 | −0.49 | −0.56 |
| Residual penetration ratio (25 °C)/% | 71.4 | 76.8 | 81.2 | 75.6 | 72.9 | 67.5 |
| The softening point difference of asphalt after segregation experiment (25 °C) | 2.9 | 1.8 | 1.4 | 1.6 | 1.9 | 2.4 |

**Table 11.** Scores for each modified asphalt group.

| Asphalt Type | A | B | C | D | E | F |
|---|---|---|---|---|---|---|
| G*/sinδ (64 °C)/KPa | 6 | 7 | 8 | 8 | 9 | 9 |
| k (MPa) | 8 | 8 | 7 | 7 | 6 | 6 |
| Penetration index PI | 6 | 8 | 9 | 8 | 8 | 7 |
| Residual penetration ratio (25 °C)/% | 7 | 8 | 9 | 8 | 7 | 6 |
| The softening point difference of asphalt after segregation experiment (25 °C) | 6 | 8 | 9 | 8 | 8 | 7 |

### 3.5.3. Weight Determination Based on Comparison Matrix Method

In the AHP, the weight of each element is established using the comparison matrix method. This method helps to resolve fuzzy concepts and determine the relative importance of each element in a hierarchical structure. First, the five evaluation elements are arranged into a 5 × 5 matrix. By comparing the elements, the values in the matrix are

determined according to the importance of each element. Then, the maximum eigenvalue of the matrix and the corresponding maximum eigenvector are calculated. Finally, the consistency test is performed. If the consistency test is passed, the maximum feature vector is considered as the weight vector. We first establish the domain of evaluation elements R, R = ($r_1$, $r_2$, $r_3$, $r_4$, $r_5$), and select two elements $r_i$ (i = 1, 2, 3, 4, 5) and $r_j$ (j = 1, 2, 3, 4, 5) to compare the importance of the two elements; $r_{ij}$ represents the judgment value of the importance of element $r_i$ to element $r_j$. The correspondence between the importance and the actual value is shown in Table 12.

**Table 12.** Judgment matrix scale and its definition.

| Importance ($r_i$ vs. $r_j$) | Judgement Matrix Scale $r_{ij}$ (5/5-9/1 Scale) |
|---|---|
| Equally important (level 0) | 5/5 = 1 |
| Slightly important (Level 1) | 6/4 = 1.5 |
| Important (Level 2) | 7/3 = 2.33 |
| Very important (Level 3) | 8/2 = 4 |
| Absolutely important (Level 4) | 9/1 = 9 |
| Intermediate state (-) | 5.5/4.5 = 1.222  6.5/3.5 = 1.875 7.5/2.5 = 5  8.5/1.5 = 5.667 |

The comparison matrix model is shown in Table 13. This study chooses high temperature performance as the benchmark. In addition to high temperature performance, the importance of other factors needs to be calculated according to high temperature performance. The following matrix is established by 5 × 5 $r_{ij}$, $r_{ij}$ (i = 1, 2, 3, 4, 5; j = 1, 2, 3, 4, 5):

$$R = \begin{pmatrix} r_{11} & r_{12} & r_{13} & r_{14} & r_{15} \\ r_{21} & r_{22} & r_{23} & r_{24} & r_{25} \\ r_{31} & r_{32} & r_{33} & r_{34} & r_{35} \\ r_{41} & r_{42} & r_{43} & r_{44} & r_{45} \\ r_{51} & r_{52} & r_{53} & r_{54} & r_{55} \end{pmatrix}$$

**Table 13.** Comparison of matrix models.

| Index | High Temperature Performance | Low Temperature Performance | Aging Resistance | Temperature Susceptibility | Storage Stability |
|---|---|---|---|---|---|
| High temperature performance | 1 | Importance of high temperature performance/low temperature | Importance of high temperature performance/aging resistance | Importance of high temperature performance /temperature susceptibility | Importance of high temperature performance/Storage stability |
| Low temperature performance | | 1 | Importance of low temperature performance /aging resistance | Importance of low temperature performance /temperature susceptibility | Importance of low temperature performance /Storage stability |
| Aging resistance | | | 1 | Importance of temperature susceptibility/Storage stability | Importance of aging resistance/Storage stability |
| Temperature susceptibility | | | | 1 | Importance of temperature susceptibility/Storage stability |
| Storage stability | | | | | 1 |

To calculate the weightage of each performance, Northeast China has been chosen as the reference region. Utilizing the climate zoning-temperature map of asphalt pavement in China, the high and low temperature grades of each climate zone can be determined, allowing for the estimation of temperature difference ranges in each zone. Based on these temperature difference ranges, each climate zone can be roughly classified into three grades, denoted by labels 1, 2, and 3. Prior research has identified temperature and ultraviolet exposure as the primary factors influencing asphalt pavement aging. Consequently, China's annual solar radiation distribution map divides the country's radiation intensity into three grades, also represented by labels 1, 2, and 3. A smaller label indicates more challenging climatic conditions. The climate factor levels of the two sub-regions in Northeast China are shown in Table 14:

**Table 14.** Grade of climate factors in Northeast China.

| Climate Zoning | High Temperature Grade | Low Temperature Grade | Temperature Difference Grade | Ultraviolet Radiation Intensity | Coupling Grade of High Temperature and Ultraviolet Radiation Intensity |
|---|---|---|---|---|---|
| 2-1 partition | 2 | 1 | 1 | 2 | 2 |
| 2-2 partition | 2 | 2 | 2 | 3 | 3 |

Establish each partition judgment matrix:

$$\text{2-1 partition:} \begin{pmatrix} 1 & 1/1.5 & 1 & 1/1.5 & 1.5 \\ 1.5 & 1 & 1 & 1 & 1.5 \\ 1 & 1 & 1 & 1.5 & 1 \\ 1.5 & 1 & 1/1.5 & 1 & 1 \\ 1/1.5 & 1/1.5 & 1 & 1 & 1 \end{pmatrix}$$

$$\text{2-2 partition:} \begin{pmatrix} 1 & 1 & 1.5 & 1 & 1.5 \\ 1 & 1 & 1.5 & 1 & 1.5 \\ 1/1.5 & 1/1.5 & 1 & 1/1.5 & 1 \\ 1 & 1 & 1.5 & 1 & 1 \\ 1/1.5 & 1/1.5 & 1 & 1 & 1 \end{pmatrix}$$

The feature vector is as follows:

2-1 partition: $W_{2-1} = (0.410, 0.513, 0.479, 0.445, 0.375)^T$

2-2 partition: $W_{2-2} = (0.511, 0.511, 0.340, 0.474, 0.372)^T$

### 3.5.4. Weight Consistency Test of Judgment Matrix

In order to enhance the reliability of the evaluation results, it is imperative to evaluate the consistency of the calculation results derived from the judgment matrix. The consistency ratio (CR) is utilized as a criterion for evaluating the consistency of the judgment matrix.

Typically, a CR value less than 0.1 is considered acceptable for achieving satisfactory consistency. If the calculated CR value exceeds 0.1, it indicates a certain level of inconsistency in the judgment matrix, which may lead to less reliable evaluation results. The specific calculation formula is as following Formulas (5) and (6):

$$CI = \frac{\lambda_{max} - n}{n - 1} \tag{5}$$

$$CR = \frac{CI}{RI} \tag{6}$$

where *CI* is the matrix consistency index, $\lambda_{max}$ is the maximum eigenvalue of the judgment matrix, *n* is the order of the judgment matrix, the *RI* value can be obtained by looking up Table 15, and the *RI* value corresponding to the fifth-order matrix is 1.12.

**Table 15.** Random consistency RI.

| n Matrix | 3 | 4 | 5 | 6 | 7 |
|---|---|---|---|---|---|
| RI | 0.52 | 0.89 | 1.12 | 1.26 | 1.36 |

It is known by calculation,

2-1 partition: $\lambda_{max}$ = 5.0996 CI$_{2-1}$ = 0.0249 CR$_{2-1}$ = $\frac{CI_{2-1}}{RI}$ = 0.0222 < 0.1

2-2 partition: $\lambda_{max}$ = 5.0198 CI$_{2-2}$ = 0.0050 CR$_{2-2}$ = $\frac{CI_{2-2}}{RI}$ = 0.0044 < 0.1

In general, the smaller the CR value is, the better the consistency of the judgment matrix is. In general, the CR value is less than 0.1, and the judgment matrix meets the consistency test. Through calculation, the CR calculation results of each partition are less than 0.1, which shows that the matrix has high reliability. The calculation results of the performance weights in the two climate zones are shown in Table 16.

**Table 16.** Recommended performance weights in the different climate zones.

| Climate Zoning | High Temperature Performance | Low Temperature Performance | Aging Resistance | Temperature Susceptibility | Storage Stability |
|---|---|---|---|---|---|
| 2-1 | 0.410 | 0.513 | 0.479 | 0.445 | 0.375 |
| 2-2 | 0.511 | 0.511 | 0.340 | 0.474 | 0.372 |

### 3.5.5. Asphalt Comprehensive Performance Index Calculation

According to the scoring basis of each road performance evaluation index and the results of each performance score, combined with the weight of each road performance, the comprehensive performance index of asphalt can be obtained. The comprehensive performance index (*P*) is a calculation based on the technical index scores and weight levels assigned to evaluate the overall performance of asphalt roads. The calculation formula of *p* value is as following Formula (7):

$$P = A\omega_A + B\omega_B + C\omega_C + D\omega_D + E\omega_E \tag{7}$$

In the formula: $\omega_A, \omega_B, \omega_C, \omega_D, \omega_E$ are the weight coefficients of rutting factor, k index, penetration index PI, penetration ratio after aging, and softening point difference of segregation experiment, respectively. The results are shown in Table 17.

**Table 17.** Evaluation results of asphalt under different partitions.

| Climate Zoning | A | B | C | D | E | F |
|---|---|---|---|---|---|---|
| 2-1 | 14.803 | 17.366 | 18.562 | 17.263 | 16.715 | 15.416 |
| 2-2 | 14.744 | 17.153 | 18.339 | 17.153 | 16.679 | 15.493 |

Based on the calculation results, it is evident that the comprehensive performance index of C asphalt, which is 4% SBR/0.04% GR composite modified asphalt, is the highest among both the 2-1 and 2-2 partitions. The comprehensive performance index values are 18.562 and 18.339, respectively. These values indicate a significant improvement when compared to 4% SBR modified asphalt. Therefore, considering the integration of comprehensive performance indicators, the optimal content of GR in the corresponding modified asphalt is determined to be 0.04%.

## 4. Conclusions

In this study, through three major index tests, DSR test, BBR test, and MSCR test, infrared spectrum test and fluorescence microscope test, combined with analytic hierarchy process, a variety of technical properties of composite modified asphalt were comprehensively analyzed. Finally, the optimal content of graphene in modified asphalt and its

modification mechanism were determined in a quantifiable form, and the corresponding modification effects were analyzed. The conclusions are as follows:

(1) On the basis of adding 4 % SBR, the addition of GR can significantly improve the high temperature rutting resistance of modified asphalt, reduce the temperature sensitivity of modified asphalt, improve the temperature sensitivity of asphalt, and improve the storage stability of asphalt, but it will have a weak impact on the low temperature performance of asphalt. The technical indices of GR/SBR composite modified asphalt demonstrate substantial enhancements when compared to SBR modified asphalt. These improvements broaden the practical application range of these materials in road engineering, providing engineers and practitioners with more versatile and reliable options for various road construction projects.

(2) It can be known from the infrared spectrum that the incorporation of graphene did not produce a new absorption peak in the figure. Through the observation of fluorescence microscope images, GR molecules are dispersed in asphalt and SBR particles in a small size, and some GR are in contact with the SBR interface. The dispersed phase is more uniform and forms a stable structure, which improves the stability of asphalt binders, thereby reducing the impact of high temperature and aging on asphalt and polymers.

(3) For the first time, a comprehensive evaluation index of asphalt is proposed, which comprehensively considers factors such as low temperature k index, high temperature rut factor, penetration index PI, short-term aging residual penetration, and softening point difference of asphalt after segregation experiment. At the same time, the analytic hierarchy process is used to establish the optimization system of modified asphalt weight factors. Considering the environmental and climatic conditions in Northeast China, the comprehensive performance of asphalt with 0.04% GR and 4% SBR stands out as the most prominent. This composite modification exhibits notable improvements when compared to SBR modified asphalt. The enhanced performance of this modified asphalt formulation offers advantages to road builders, providing them with a convenient and effective solution in their decision-making processes. By incorporating GR and SBR, the asphalt can better withstand the specific challenges posed by the region's environmental and climatic characteristics, leading to improved road performance and durability.

Based on the aforementioned test results and analysis, the incorporation of an optimal amount of ground rubber (GR) demonstrates enhanced road performance of asphalt in comparison to SBR modified asphalt. This improvement not only expands the application scope of modified asphalt in road engineering but also enhances its overall effectiveness. For the first time, we have introduced a comprehensive evaluation index for asphalt that considers various factors such as the low-temperature k-value, high-temperature rut factor, penetration index (PI), residual penetration after short-term aging, and softening point difference after segregation experiment. This comprehensive evaluation allows for a holistic assessment of asphalt performance. Furthermore, we employed the Analytic Hierarchy Process (AHP) to quantitatively determine the optimal dosage of graphene. This approach provides a systematic method for selecting the appropriate material dosage in future applications. The combination of macro and micro analysis of modified asphalt performance can make the test conclusion more accurate. The experimental results show that the appropriate amount of graphene incorporation can improve the road performance of asphalt and expand the application of modified asphalt in road engineering. This is also of great significance for the further application of nanomaterials in road engineering.

**Author Contributions:** Conceptualization, L.W.; methodology, F.L.; software, F.L.; resources, Q.Z. data curation, F.L.; writing—original draft preparation, F.L.; writing—review and editing, F.L. supervision, L.W.; project administration, Z.L. All authors have read and agreed to the published version of the manuscript.

**Funding:** This research received no external funding.

**Institutional Review Board Statement:** Not applicable.

**Informed Consent Statement:** Not applicable.

**Data Availability Statement:** Not applicable.

**Conflicts of Interest:** The authors declare no conflict of interest.

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
