# Peer review of "Investigation on the Rheological Properties and Microscopic Characteristics of Graphene and SBR Composite Modified Asphalt"

_coatings, doi:10.3390/coatings13071279_

Round 1

Reviewer 1 Report

1.      This paper discusses on graphene and SBR composite modified asphalt for engineering applications.

2.      In the abstract it is mentioned that inclusion of graphene substantially enhances the high- temperature rutting resistance of asphalt, reduces the temperature sensitivity of modified asphalt, and improves its storage stability. What were the improvements in the sated properties when compared to base samples? These findings of the work are to be expressed in quantitative terms, rather than expressing them in qualitative terms.

3.      In the Introduction section, it is highlighted that there is limited research on composite modified asphalt using graphene and SBR. Why graphene and SBR were chosen for the study? What is the novelty of the work ??.  How the uniform mixing of graphene is ensured??

4.      In section 2 on Materials and Methods, SEM image of Graphene (GR) to be provided. The Grain size of the Graphene is micro or nano? Further test standards used (ASTM) for Storage Stability, Dynamic Shear, Creep and Infrared Spectrum Test to be provided.

5.      Results and discussion section need co-relation of the results presented to the published literature.

6.      Resolution of Figures 7 and 8 are be improved.  

7.      Conclusions section need to highlight on the important findings of the work reported.

8.      The manuscript needs careful corrections for the English grammar and typo errors.

Minor editing of English language is required

Reviewer 2 Report

ARTICLE REVIEW

Manuscript ID: coatings-2496181

 Title: “Investigation on the rheological properties and microscopic characteristics of graphene and SBR composite modified asphalt”

 Authors: Lijun Wang, Fengxiang Liang, Zixia Li , Qiang Zhao

 The article is submitted for publication in the journal “Coatings”

Special Issue: “Asphalt Pavement: Materials, Design and Characterization”

There are many problems in the physical study of the composition of pavement materials. They are associated with new technologies in the development of highly durable asphalt materials. One of the important problems is the direction of studying the rheological and microscopic properties of asphalt as a result of adding graphene to bitumen.

In the article under consideration, the analysis of previously used technological methods, methods and modifications for the study of mechanical thermal and adhesive properties of new types of asphalt is carried out.

The article is written in good technical language. Figures and captions to them do not cause any remarks. References to publications and literature citations are given correctly.

No errors were found in the article. There are no remarks about the style of writing the article.

In general, the article is a completed study and is recommended for publication in the journal “COATINGS”

Author Response

We thank the reviewer for reading our paper carefully and giving the positive comments. We gratefully appreciate for your valuable suggestion. We have modified some grammar problems in the article. Thank you for reading and reviewing our Manuscript.

Reviewer 3 Report

The article by Wang L. et al. describes the modification of SBR-containing asphalt binder with graphene particles. The authors add different contents of graphene to the binder and investigate how it affects its rheological characteristics and, through them, its resistance to rutting and cracking. Then the authors optimize the binder composition for specific temperature zones in northern China using an original methodology. In general, the article is well written and contains interesting data but should be revised before publication.

Specific comments are as follows.

Lines 34-76: The introduction to the article contains very low references on such an extensive topic as the modification of asphalt with nanoparticles. Only 16 references are very weak and do not reflect the relevance and demand of this topic.

Line 39: “become one of the most widely used 39 road pavements in China.” Asphalt is widely used all over the world, not just in China.

Line 50: “However, its high temperature performance is not as good as the low temperature performance, which also greatly limits the promotion of SBR modified asphalt [5].” I disagree with this statement because it is not entirely correct. Rubber modifier in the form of styrene-butadiene copolymer or crumb rubber can increase the high-temperature properties well, but it requires a large amount of these modifiers, which greatly increases the viscosity of the road binder (see 10.1134/S1061933X1404005X).

Line 63: Therefore, asphalt is usually harder after adding nanomaterial [9]. There are many more works in which nanoparticles of silica, montmorillonite, nanocellulose, carbon nanotubes, and other are used, and where this effect is shown.

Line 71: “As a new type of nanomaterial, the application of graphene in pavement engineering has attracted more and more attention [14].” Graphene as an asphalt modifier has been considered before in other works (e.g., 10.3390/ma14092434). The introduction should be enriched with references.

Line 100: “Its physical mechanism performance, processing performance and product performance are close to natural rubber.” This phrase is incorrect. Natural rubber is polyisoprene, which has little in common with SBR except that both these polymers are in a rubbery state, and this can be said about any rubber.

Line 101: “It has the characteristics of wear resistance”. This phrase is incorrect. SBR has no wear resistance since it is a rubber with a low elastic modulus and therefore wears a lot. Perhaps the authors meant that SBR-modified asphalt has good wear resistance, and then it should be written about asphalt rather than about SBR alone.

Table 1. The table should indicate the molecular weight of styrene-butadiene rubber. In addition, the text should indicate whether SBR is a diblock or triblock copolymer and whether it has a linear or branched structure.

Line 114: “The addition of trace graphene to the polymer can change the microstructure of the asphalt binder, so that the asphalt has excellent mechanical, thermal and adhesion properties.” A reference is needed to support this statement.

Line 166. A replacement is needed: “the rotation frequency” -> “the angular frequency”.

Figures 5 and 6: It is necessary to plot error bars showing the standard deviations of the values.

Figures 7 and 8: The X and Y captions are hard to see because they are rotated so much.

Line 335: “We can also see that as the experimental temperature increases, the Jnr of each asphalt increases nonlinearly. As the temperature increases, the slope of the curve increases significantly.” Perhaps if the authors plot the dependence of log??? on temperature (or on 1/T), the dependence will be linear.

Line 420: “Infrared spectroscopy is one of the most widely used methods to study the chemical structure of polymers [25].” Infrared spectroscopy is widely used to study the chemical structure of not only polymers but also bitumens, including asphalt (see 10.3390/molecules28052065).

Line 631: “It can be seen from the infrared spectrum that the incorporation of graphene did not produce a new absorption peak in the figure, indicating that the incorporation of graphene did not react with the asphalt, and the modification of the asphalt was physical modification.” This is a speculative statement. The concentration of graphene is too small (only 0.1% in maximum) for its strong interaction with asphalt or SBR to affect the IR spectra.

The English language is good.

Round 2

Reviewer 3 Report

The authors have made the necessary corrections to the manuscript for its publication.

The English language is good.